# Effect of Dietary Magnesium Content on Intestinal Microbiota of Rats

**DOI:** 10.3390/nu12092889

**Published:** 2020-09-22

**Authors:** Arantxa García-Legorreta, Luis Alfonso Soriano-Pérez, Aline Mariana Flores-Buendía, Omar Noel Medina-Campos, Lilia G. Noriega, Omar Granados-Portillo, Rafael Nambo-Venegas, Armando R. Tovar, Alfredo Mendoza-Vargas, Diana Barrera-Oviedo, José Pedraza-Chaverri, Berenice Palacios-González

**Affiliations:** 1Unidad de Vinculación Científica de la Facultad de Medicina, de la Universidad Nacional Autónoma de Mexico en el Instituto Nacional de Medicina Genómica, Mexico City 14600, Mexico; legorretaara@gmail.com; 2Facultad de Química, Departamento de Biología, Laboratorio F-315, Universidad Nacional Autónoma de Mexico, Mexico City 04510, Mexico; lsoriano.jr@gmail.com (L.A.S.-P.); alinefloresbuendia@gmail.com (A.M.F.-B.); omarnoelmedina@gmail.com (O.N.M.-C.); pedraza@unam.mx (J.P.-C.); 3Departamento de Fisiología de la Nutrición, Instituto Nacional de Ciencias Médicas y Nutrición “Salvador Zubirán”, Mexico City 14080, Mexico; lgnoriegal@gmail.com (L.G.N.); ograpo@yahoo.com (O.G.-P.); tovar.ar@gmail.com (A.R.T.); 4Laboratorio de Bioquímica de Enfermedades Crónicas, Instituto Nacional de Medicina Genómica, Mexico City 14600, Mexico; rafaelnambo@yahoo.com.mx; 5Unidad de Secuenciación, Instituto Nacional de Medicina Genómica, Mexico City 14600, Mexico; amendoza@inmegen.gob.mx; 6Departamento de Farmacología, Facultad de Medicina, Universidad Nacional Autónoma de México, Mexico City 04510, Mexico; dianabarrera@hotmail.com

**Keywords:** dietary magnesium, minerals, gut microbiota, Wistar male rat

## Abstract

Background: Magnesium is a mineral that modulates several physiological processes. However, its relationship with intestinal microbiota has been scarcely studied. Therefore, this study aimed to assess the role of dietary magnesium content to modulate the intestinal microbiota of Wistar male rats. Methods: Rats were randomly assigned one of three diets: a control diet (C-Mg; 1000 mg/kg), a low magnesium content diet (L-Mg; 60 mg/kg), and a high magnesium content diet (H-Mg; 6000 mg/kg), for two weeks. After treatment, fecal samples were collected. Microbiota composition was assessed by sequencing the V3–V4 hypervariable region. Results: The C-Mg and L-Mg groups had more diversity than H-Mg group. CF231, SMB53, *Dorea*, *Lactobacillus* and *Turibacter* were enriched in the L-Mg group. In contrast, the phyla *Proteobacteria*, *Parabacteroides*, *Butyricimonas*, and *Victivallis* were overrepresented in the H-Mg group. PICRUSt analysis indicated that fecal microbiota of the L-Mg group were encoded with an increased abundance of metabolic pathways involving carbohydrate metabolism and butanoate metabolism. Conclusion: Dietary magnesium supplementation can result in intestinal dysbiosis development in a situation where there is no magnesium deficiency. Conversely, low dietary magnesium consumption is associated with microbiota with a higher capacity to harvest energy from the diet.

## 1. Introduction

Even though they are required in smaller quantities than macronutrients, vitamins and minerals are crucial to maintaining normal metabolic function. An increasing number of individuals have micronutrient deficiencies, particularly people with insufficient fresh fruit and vegetable consumption [1]. Among these micronutrients, magnesium is a mineral that plays an essential role in several cellular reactions, including DNA repair and replication, mineral transport (calcium and potassium), and metabolism [2]. Magnesium deficiency results in impaired in psychiatric disorders, hypertension, and insulin resistance, among others. It has also been reported that magnesium prevents infections by enhancing immunity and decreasing pro-inflammatory cytokines [2,3,4,5].

Vitamins and minerals are associated with bacterial physiological processes reshaping the intestinal microbiota’s structure and composition [6,7,8]. For instance, zinc deficiency in piglets is associated with a lower abundance of *Clostridiales* and *Verrucomicrobia* and a higher abundance of *Enterobacteriaceae* and *Enterococcus* [8]. Interestingly, although zinc supplementation could mitigate this effect on intestinal microbiota, studies have shown that excess zinc can cause adverse effects. Zackular et al. demonstrated that excess dietary zinc increased *Clostridium difficile* disease by enhancing toxin activity, modifying the host’s immune response, and promoting *Enterococcus*, *Porphorymonadaceae*, *Lachnospiraceae*, and *Clostridia* cluster XI expansion [9]. Furthermore, dietary iron modifies the structure of intestinal microbiota. Constante et al. showed that an iron-rich diet reduced microbial diversity, decreased *Firmicutes*, and increased *Proteobacteria* [10].

The above information suggests that mineral concentrations in the host must be in strict homeostasis since, as they are essential for humans, many bacteria need them for their survival and biological processes and compete with the host to obtain them [8]. Pathogenic bacteria can take advantage of both an excess and a deficiency of a mineral to mitigate and alleviate the normal mechanisms that the host performs to retain the inorganic compounds it needs [10]. Moreover, mineral availability could also favor or inhibit commensal bacteria’s expansion in the intestine [9]. Hence, the control of the availability of minerals by the host strengthens the fact that the distribution and concentration of minerals serve as critical contributors to host–pathogen communications and commensal bacteria, which underlies the need for a rigorous balance [8], which is, in part, determined by dietary intake of minerals. Nevertheless, regarding magnesium, little is known about how this mineral modifies intestinal microbiota composition. Therefore, the present study aims to assess dietary magnesium content’s effect on the intestinal microbiota of male Sprague Dawley rats.

## 2. Materials and Methods

### 2.1. Animals

The Animal Committee approved the study of the Chemistry Faculty, Universidad Nacional Autónoma de México (UNAM), Mexico City, on 20 September 2019 (CICUAL/384-2/19). Animals were obtained from the Faculty of Medicine, UNAM. Male Wistar rats aged 7 weeks were kept in individual cages at a controlled room temperature with free access to water and food and 12 h light/dark cycles. The rats were randomly assigned into three groups: six rats were fed a control diet (C-Mg; 1000 mg/kg), six rats were fed a low magnesium content diet (L-Mg; 60 mg/kg), and five rats were fed a high magnesium content diet (H-Mg; 6000 mg/kg), for two weeks each. The rats were euthanized after 2 weeks of treatment in consideration of the effects that magnesium deficiency has in about one week [11], and the fact that rats developed diarrhea as a consequence of the magnesium high diet treatment. Diet composition was based on the recommendations of the AIN-93 [12], containing 19.85% g/kg of protein (casein), 65% g/kg of carbohydrates (sucrose and starch), and 5% g/kg of fat (soy oil). A mineral mix without MgO (TD.180705, Envigo, Indianapolis, IN, USA) was used to modulate magnesium concentration by adding MgO (243388, Sigma-Aldrich, Germany) to 60, 1000, or 6000 mg/kg, based on the experiments of Rayssiguier et al. [11] and Martin et al. (2008) [13]. Weight and food consumption were determined every other day during the protocol. After two weeks, each rat’s stool was collected and stored at −70 °C until analysis. At the end of the experiment, the rats were killed by decapitation after anesthetizing with sodium pentobarbital (50 mg/Kg, i.p.). Blood was centrifugated at 1500× *g* for ten minutes to obtain serum. The serum was stored at −70 °C until analysis.

### 2.2. Biochemical Parameters

Serum glucose, total cholesterol, and triglycerides were evaluated with a COBAS C111 (Roche, Basel, Switzerland).

### 2.3. Bacterial DNA Purification, Amplification, and Sequencing

Fresh feces samples were collected in a sterile polypropylene container, frozen, and kept at −70 °C until processing. Later, bacterial DNA was obtained by DNA stool kit (QIAGEN), following the instructions of the manufacturer. A spectrophotometer was employed to determine DNA concentration. Specific primers were used for the sequencing of the V3–V4 hypervariable region. DNA libraries were sequenced at the Unidad de Secuenciación at INMEGEN by an Illumina Miseq 2 × 300 platform (Illumina, San Diego, CA, USA). Processing of the Illumina FASTQ reads was performed using the quantitative insights into microbial ecology (QIIME 1.8) software package [14]. The UCHIME algorithm was used for detection and removal of chimeric sequences [14]. An alignment to the Greengenes database was performed using some representative sequences (sequences are available in https://www.ncbi.nlm.nih.gov/sra/PRJNA649973).

### 2.4. Statistical Analysis

A one-way ANOVA test, followed by Bonferroni post-hoc analysis, was used to evaluate the parametric variables. The Kruskal–Wallis test with Tukey’s multiple comparisons was used to determine significant variables among the groups. An LEfSe method was performed for assessment of the microbial communities’ differences (LDA < 2). For microbial community metagenome prediction with PICRUSt, the sequences were aligned against Greengenes, and Operational Taxonomic Units (OTUs) were assigned at 97% identity. The OTU table was used for microbial community metagenome prediction on the Galaxy interface (https://huttenhower.sph.harvard.edu/galaxy) and the web-based tool MicrobiomeAnalyst (http://www.microbiomeanalyst.ca). To evaluate the association and contribution of the variables to differentiate L-Mg and H-Mg group from the C-Mg group, an analysis of random forest decision trees was performed according to mean accuracy.

## 3. Results

### 3.1. Body and Biochemical Parameters

In the present study, anthropometric and biochemical parameters from the three intervention groups were analyzed. No significant differences among the weight, weight gain, serum glucose, triglyceride, and cholesterol levels were observed (Table 1).

### 3.2. Intestinal Microbiota Characterization

The amplicon of the V3–V4 region of 16S rRNA gene was analyzed by sequencing (paired ends of 300 bp). An average of 22,474 sequences per sample were generated. Chao1, Shannon index, and observed species were used for the analysis of alpha diversity (Figure 1a–c). Regarding alpha diversity, there were significant differences among the groups. The C-Mg and L-Mg groups had more diversity, richness, and observed species than the H-Mg group (*p* = 0.0019, *p* = 0.061, and *p* = 0.0006, respectively). No differences were found between C-Mg and L-Mg (Chao *p* = 0.898, Shannon index *p* = 0.787, Observed species *p* = 0.188). To compare the microbial communities’ compositions, we calculated the beta diversity by UniFrac distances (Figure 1d). Differences in community compositions were observed between the H-Mg group and C-Mg and L-Mg groups (*p* = 0.001). No significant difference was found between the C-Mg and L-Mg groups (*p* = 0.211) or between the C-Mg and L-Mg groups (*p* = 0.211).

A total of three phyla dominated the microbiota composition, the predominant phyla were *Firmicutes*, with a mean abundance of 59% across the whole dataset, followed in rank abundance order by *Bacteroidetes*, accounting for 21%, and *Proteobacteria,* accounting for 18%. In contrast, *Actinobacteria*, *Tenericutes*, *Verrucomicrobia*, and others accounted for 2% (Figure 2a). To assess differences at any taxonomic levels among each group (Figure 2b), we performed an analysis using the LEfSe algorithm. Genus CF231, SMB53, *Dorea*, *Lactobacillus* and *Turibacter* were increased in the L-Mg group (LDA score ≥ 4). In contrast, *Desulfovibrio* (genus), *Parabacteroides* (genus), *Helicobacter* (genus), *Butyricimonas* (genus), *Sutterella* (genus), *Campylobacter* (genus), *Mycoplasma* (genus), and *Victivallis* (genus) were overrepresented in the H-Mg group (Figure 2c).

### 3.3. Co-Occurrence and Co-Exclusion Patterns

Correlation networks are useful to identify potential interactions between bacteria that could represent mutualistic, commensal, parasitic, or even competitive relationships. To obtain a measurement of association between OTUs and magnesium consumption, we inferred SparCC correlation coefficients. We identified 67 associations that had a *p*-value less than 0.05, 100 permutations and *r* value ≥ 6; 39 were positive (*r* ≥ 0.5), and 28 were negative (*r* ≤ −0.5) from over 160 relationships that were observed in total (Figure 3). A co-exclusion pattern (gray lines) was observed between the abundant taxa of the H-Mg group and the representative taxa of the L-Mg group. For instance, *Desulfovibrio*, *Parabacteroides*, *Helicobacter*, *Victivallis*, *Sutterella* and *Campylobacter* showed a negative correlation with SMB53, CF231, *Dorea,* and *Prevotella*. Meanwhile, a co-occurrence pattern (orange lines) was observed exclusively within each group. For example, representative taxa of the H-Mg, *Butyricimonas*, *Desulfovibrio*, *Parabacteroides*, *Helicobacter*, *Victivallis*, *Sutterella, Anaerotruncus*, and *Mucispirillum* exhibited the strongest positive correlation among them, whereas the representative taxa of the L-Mg, CF231, SMB53, *Dorea*, *Prevotella*, *Phascolarcbacterium*, *Allobaculum* and *Coprococcus*, rc4_4, and p75_a5 exhibited a positive correlation among them. Finally, two hubs (nodes connected to most other nodes) were identified; the first hub was *Desulfovibrio,* a representative taxa of the H-Mg group, and the second hub was *Dorea* and *Prevotella,* as representative taxas of the L-Mg group and C-Mg group, respectively.

### 3.4. Effect of Dietary Magnesium Content on Microbial Functional Pathways

PICRUSt was utilized to infer functional differences in gut microbiota. At the highest KEGG (Kyoto Encyclopedia of Genes and Genomes) pathway hierarchy level, several differences in bacterial metabolic pathways among groups were found (Figure 4a). The L-Mg microbiome was enriched for KEGG pathways for Phe, Tyr, and Trp biosynthesis and starch/sucrose, galactose, glycerolipid, and butanoate metabolism; meanwhile, the H-Mg microbiome was enriched for lipopolysaccharide biosynthesis (Figure 4b). To evaluate the association and contribution of the variables to differentiate the L-Mg and H-Mg groups from the C-Mg group, an analysis of random forest decision trees was performed according to mean accuracy. The features that allowed discrimination between L-Mg, C-Mg and H-Mg were: K02587; nitrogenase molybdenum-cofactor synthesis protein, K02774; galactitol PTS system EIIB component and K01575; acetolactate decarboxylase (Figure 4c).

## 4. Discussion

Dietary patterns constitute the daily consumption of certain foods and beverages, which results in a specific blend of macro and micronutrients. One of the dietary patterns that has given rise to concern is the increase in the consumption of saturated fats, animal foods, and refined sugars, added to the decreased intake of fruits, vegetables, and whole grains, which results in a diet deficient in fiber and micronutrients, such as vitamins and minerals, which negatively impacts the composition of gut microbiota [15].

Specifically, magnesium deficiency is associated with systemic and intestinal inflammation, although the mechanisms of these findings have not yet been fully described [16]. Interestingly, the present study demonstrates that dietary magnesium content affects the structure, composition, and functional pathways of rats’ intestinal microbiota in two weeks. Mainly, high dietary magnesium decreased community diversity, while low dietary magnesium did not modify diversity. Our results contrast with previous research reported by Winther et al., who demonstrated that a diet deficient in magnesium modified bacterial diversity in mice [17]. However, this study compared magnesium deficiency with a magnesium excess, lacking a group of control magnesium content. Moreover, the discrepant results could be related to the hypervariable region that was analyzed [V3 or V3–V4] [4,17,18,19] or the methods to evaluate intestinal microbiota, namely denaturation gradient gel electrophoresis [4,17], real-time quantitative PCR [18], or sequencing [4]. Additionally, the author who studied the same hypervariable region and performed the same massive sequencing as Crowley et al., suggests being cautious with the results, since the mineral mixture contained, in addition to magnesium, other trace minerals [18]. Moreover, the lower diversity observed with high dietary magnesium has also been observed in other minerals, such as zinc and iron. Zinc deficiency has been associated with the development of diarrhea [8], and excess of zinc can cause adverse effects, such as intestinal microbiota dysbiosis that will exacerbate a *Clostridium difficile* infection [9]. Regarding iron, iron-supplemented mice showed less diversity than the iron-deficient group after 27 days [20].

After 15 days, a low dietary magnesium level did not modify bacterial diversity compared to the control group. Despite no differences regarding diversity, low dietary magnesium increased CF231 and SMB53. The function of CF231 is relatively unknown. Recently, CF231 taxa were associated with improved feed efficiency in pigs [21]; the SMB53 genus, belonging to the *Clostridiaceae* family, consumes mucus and plant-derived saccharides [22]. Interestingly, SMB53, *Lactobacillus*, and *Turicibacter* are increased in mice that spontaneously develop obesity and diabetes [23]. *Turibacter* has been correlated with butyrate and the consumption of a high resistant starch diet [24]; however, the metabolism of *Turicibacter* and its interaction with the host are still not clear. Finally, higher counts of *Lactobacillus* spp. have been observed in the feces of type 2 diabetes patients when compared to control subjects [25]. Notably, we found that *Lactobacillus*, *Turicibacter* and SMB53 were overrepresented in the L-Mg group. Several studies have reported that lower magnesium consumption is correlated with an increased risk of insulin resistance [26,27,28]. It is possible that, in the long term, the taxa found in the L-Mg group will favor the development of obesity and insulin resistance. To confirm this observation, it is recommended that the intervention duration be longer and/or lower doses of magnesium be used.

It is important to point out that, in the present study, the H-Mg group presented diarrhea without affecting weight gain or animal growth. Although oral Mg has been reported to cause diarrhea in clinical studies [29,30], the experiments listed above did not report diarrhea in the animal models fed Mg supplementation diets [11,13]. We only know about one report that evaluated the effect of Mg-induced diarrhea in the gut microbiota, which used “purgative doses” of 0.5 mL saturated MgSO4 solutions (129 mg of total Mg). In our case, the average Mg supplemented diet consumption (15.7 g/rat/day) did not quite reach the Mg quantity used for the purgative dose (94.7 mg of total Mg), but it is probable that diarrhea may well have produced a change in the microbiota, as reported in the previous article [31]. Thus, it is possible that the observed changes regarding diversity and taxonomy could be attributed to diarrhea. Supporting this hypothesis, Carroll et al. have reported reduced microbial richness and increased level of *Proteobacteria* phyla (particularly the class γ-*Proteobacteria* and the family *Enterobacteriaceae*) in diarrheic patients [32]. In agreement, we found that the H-Mg group was enriched in β-, δ- and ε-*Proteobacteria*. Besides, the H-Mg group presented a significant increase of *Butyricimonas* and *Parabacteroides*. Recently, Xia et al. negatively correlated *Proteobacteria, Butyricimonas*, *Parabacteroides,* and *Bilophila* abundance with liver inflammation and intestinal epithelial claudin expression [33]. Moreover, *Victivallis* and *Butyricimonas* abundance have been positively correlated with hepatic lipid accumulation [34]. Altogether, this evidence suggests that the intestinal microbiome profile of rats fed high dietary magnesium could be potentially more pathogenic and pro-inflammatory than those from the C-Mg or L-Mg group. However, it is recommended to perform further studies to evaluate the effect of high dietary magnesium in rats with a magnesium deficiency and to use increasing doses of magnesium for supplementation.

Interestingly, in the present study, PTS and acetolactate decarboxylase were enriched in the L-Mg group. PTS is a translocation system to transports sugars, and acetolactate decarboxylase is implicated in butanoate metabolism and C5-branched dibasic acid metabolism. Furthermore, the L-Mg microbiome was enriched for sugars, glycerolipid, and butanoate metabolism. Altogether, this evidence suggests that the L-Mg microbiome could have a higher capacity to harvest energy from the diet. To corroborate this observation, the intervention duration be longer and/or lower magnesium doses be used. However, Turnbaugh et al., indicated that the ob/ob mice microbiome was enriched with enzymes involved in breaking-down sugars and butyrate metabolism [35]. Studies have shown that intestinal microbiota have a significant influence on nutrient uptake and energy utilization. Further analysis of the metabolism of these bacteria is required in low or deficient magnesium diets.

Diet affects the relative and absolute abundance of intestinal microbiota and is also capable of affecting the growth kinetics of some taxa [36], highlighting the extensive inter-species interactions in microbial communities. Phylotypes’ abundance profiles can reflect such interactions by co-occurrence and co-exclusion patterns. In the present study, co-exclusion patterns between the H-Mg hub (*Desulfovibrio*; δ-*Proteobacteria*), L-Mg hub (*Dorea*; *Clostridia*) and C-Mg (*Prevotella*; *Bacteroidia*) were observed. Interestingly, Petersen et al., demonstrated by cohousing that the bacterial genus *Desulfovibrio* could lead to decreased *Dorea* abundance. The authors point out that the loss of *Clostridia* class and the increase in *Desulfovibrio* are relevant for metabolic and inflammatory diseases since *Desulfovibrio* has been associated with hydrogen sulfide production detected in patients with type 2 diabetes, obesity, and inflammatory bowel syndrome [37].

## 5. Conclusions

In conclusion, dietary magnesium supplementation, in a situation where there is no mineral deficiency, can result in intestinal dysbiosis development. Conversely, low dietary magnesium consumption is associated with a microbiota with a higher capacity to harvest energy from the diet.

## Figures and Tables

**Figure 1 nutrients-12-02889-f001:**
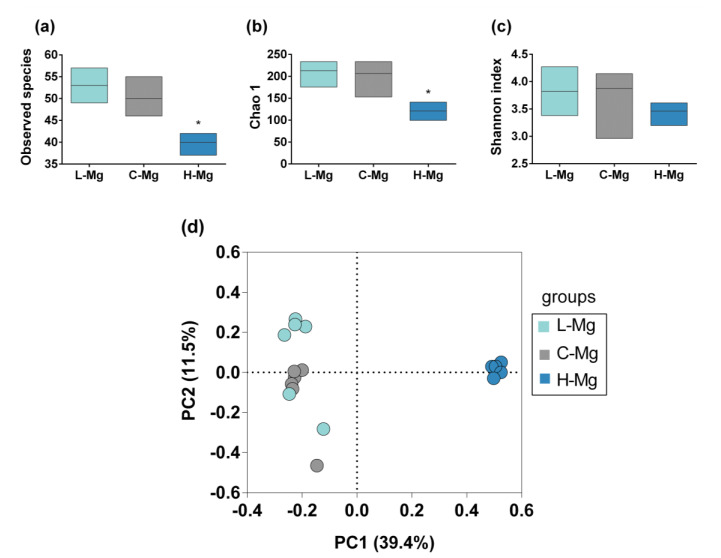
Alpha and beta diversity among the groups. (**a**) Observed species among the study groups. (**b**) Comparison of species richness among the groups. (**c**) Comparison of species diversity among the groups. (**d**) Beta diversity. Principal Coordinates Analysis (PCoA) plot based on generalized UniFrac distances among the groups. Low magnesium diet (L-Mg) (light blue circles), control diet (C-Mg) (gray circles), and high magnesium diet (H-Mg) (dark blue circles). * *p* < 0.05 compared to the C-Mg group. Boxes represent the minimum value to the maximum value, with the internal horizontal line representing the median.

**Figure 2 nutrients-12-02889-f002:**
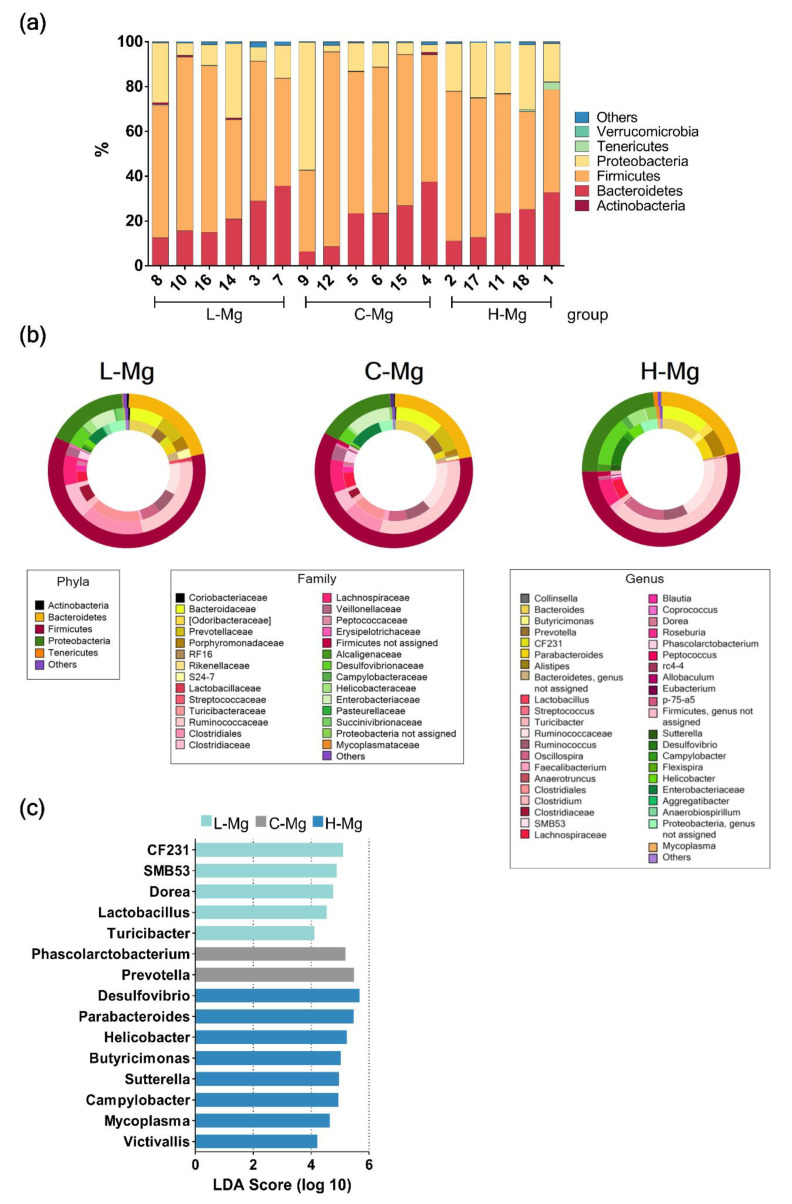
Bacterial taxonomy among the groups. (**a**) Phylum-level composition (% relative abundances) among the three study groups. (**b**) Fecal microbiota composition among the study groups. The outermost ring indicates phylum level; the middle ring displays composition at the family level and the innermost ring at genus level. (**c**) Linear discriminant analysis effect size among the study groups.

**Figure 3 nutrients-12-02889-f003:**
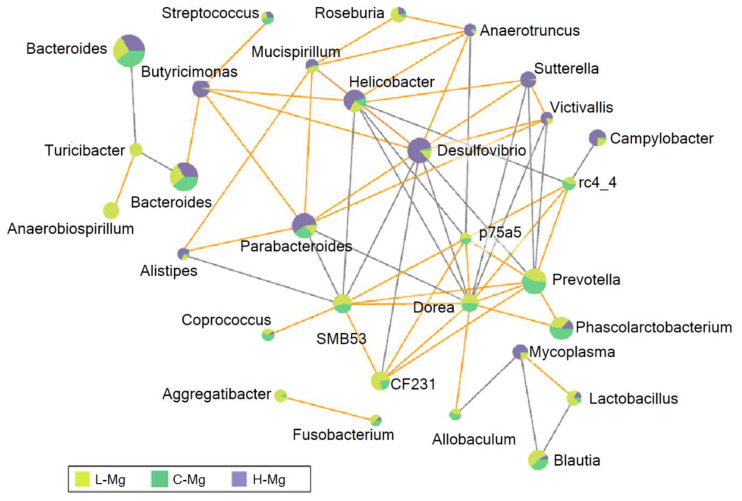
Network analysis of intestinal microbiota using SparCC correlation coefficients between the groups. The networks indicate abundant sequences at the genus level. The nodes represent genera; the edges represent the correlation between genera. The gray lines indicate negative correlations; the orange lines are for positive correlations. Nodes are colored according to their abundance among the groups.

**Figure 4 nutrients-12-02889-f004:**
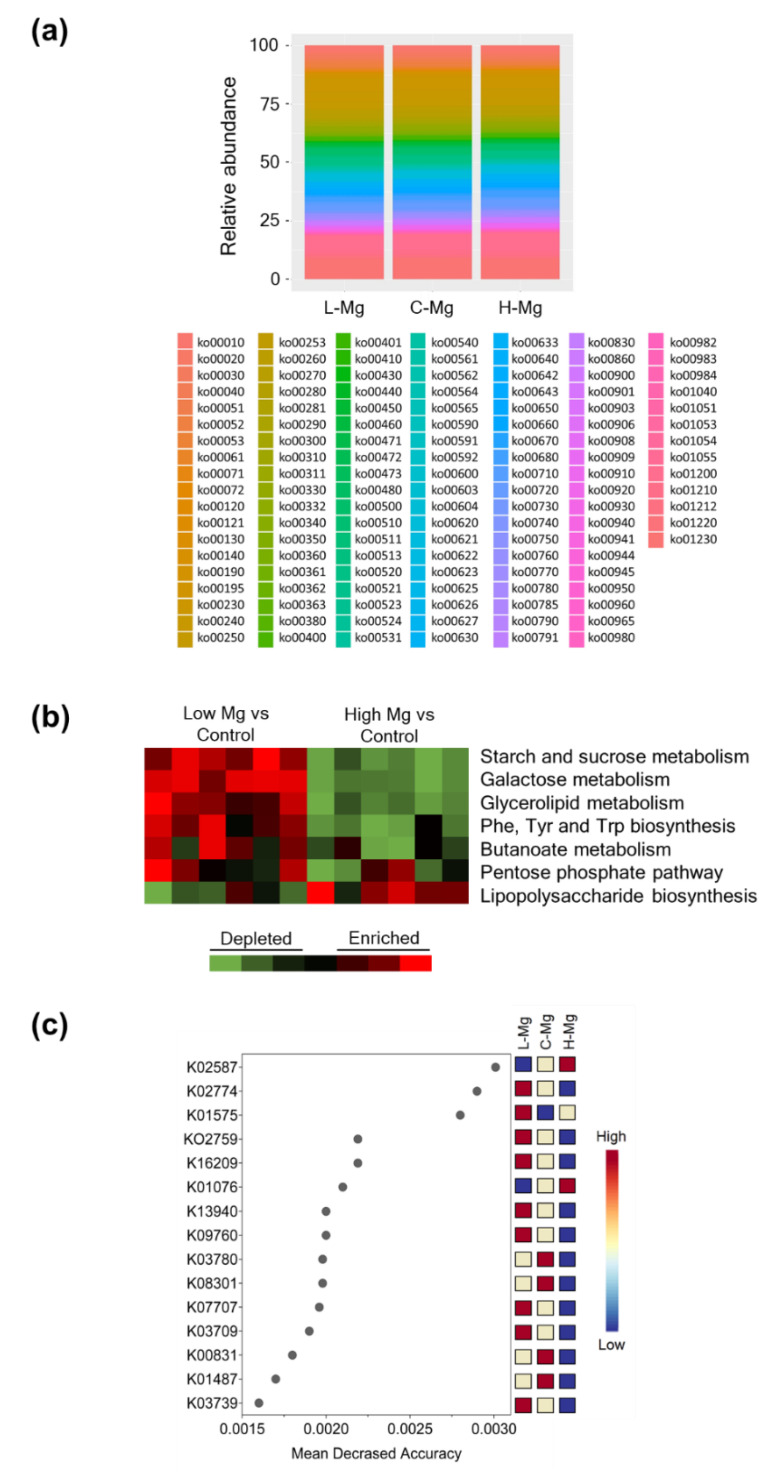
Inferred metagenomic analyses from the fecal microbiomes of the L-Mg, C-Mg, and H-Mg groups. (**a**) The relative contribution of KEGG (Kyoto Encyclopedia of Genes and Genomes) pathways encoded in the gut microbiota of the L-Mg, C-Mg, and H-Mg groups. (**b**) KEGG pathways consistently enriched or depleted in the fecal microbiomes of the L-Mg or H-Mg groups compared with the C-Mg group. Red denotes enrichment and green indicates depletion. Black indicates pathways whose representation is not significantly different. (**c**) Plot of the most significant KEGG Orthology (KO) identified by random forest. The features are ranked by the mean decrease in classification accuracy (500 permutations). Red denotes the most important feature and blue indicates the least important feature.

**Table 1 nutrients-12-02889-t001:** Body and biochemical parameters among the groups.

	Low-Mg (L-Mg)	Control	High-Mg (H-Mg)
Basal weight (g)	154 ± 4.3	154 ± 5	153.5 ± 3.9
Final weight (g)	214.7 ± 6.5	205.8 ± 12.8	210.3 ± 6.6
Weight gain (g)	60.7	56.8	66.0
Glucose (mg/dL)	123 ± 12	102 ± 7	116 ± 11
Triglycerides (mg/dL)	108 ± 17	128 ± 31	74 ± 5
Cholesterol (mg/dL)	59 ± 4	58 ± 4	58 ± 4
Food intake (g/d)	18.08 ± 0.297	16.9 ± 0.47	16.6 ± 0.37

Data are presented as the mean ± SEM. Krustal–Wallis test, *p* < 0.05.

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
