# Peer review of "Effect of Dietary Magnesium Content on Intestinal Microbiota of Rats"

_nutrients, 2020, doi:10.3390/nu12092889_

Round 1

Reviewer 1 Report

I read the manuscript with great interest. There are some areas where the authors can improve the manuscript further to make it more readable. Please find below my suggestions /comments for making it more readable.

  1. Figure 1. (d) - Not clear. Suggest to improve the quality of the figure
  2. Figure 2 (b) - Text overlapping (Phylum - under L-Mg) and again suggest to improve the quality of the figure /text 
  3. Figure 3 - suggest to improve the quality of the figure
  4. Figure 4 (a) - suggest to improve the quality of figure
  5. Line 236-237 - "Specifically, magnesium deficiency is associated with systemic and intestinal inflammation, although the mechanisms of these findings have not yet been fully described" -  Reference missing
  6. Lines 237-240 - Complex sentence - suggest to modify to improve readability
  7. Lines 251-253 - "...and excess of zinc can cause adverse effects, such as intestinal microbiota dysbiosis that will exacerbate a Clostridium difficile infection" - Reference missing
  8. Line 258 - 'fed efficiency' - e missing (feed efficiency).

Author Response

Reviewer: 1

We thank the reviewers for their valuable comments and suggestions, which have strengthened the revised manuscript substantially. Our point-by-point response to their concerns is provided below.

1. Figure 1. (d) - Not clear. Suggest improving the quality of the figure

This was done as requested

2. Figure 2 (b) - Text overlapping (Phylum - under L-Mg) and again suggest to improve the quality of the figure /text

This was done as requested

3. Figure 3 - suggest to improve the quality of the figure

This was done as requested

4. Figure 4 (a) - suggest to improve the quality of figure

This was done as requested

5. Line 236-237 - "Specifically, magnesium deficiency is associated with systemic and intestinal inflammation, although the mechanisms of these findings have not yet been fully described" -  Reference missing

This was done as requested: “Specifically, magnesium deficiency is associated with systemic and intestinal inflammation, although the mechanisms of these findings have not yet been fully described [15; Trapani V, Petito V, Di Agostini A, Arduini D, Hamersma W, Pietropaolo G, et al. Dietary Magnesium Alleviates Experimental Murine Colitis Through Upregulation of the Transient Receptor Potential Melastatin 6 Channel. Inflamm Bowel Dis. 2018;24(10):2198-2210.].”

6.Lines 237-240 - Complex sentence - suggest to modify to improve readability

This was changed as requested. The sentence now reads: “Interestingly, Crowley et al., found that consumption of marine mixture increased the bacterial diversity in gut microbiota; however, the authors suggested being cautious with the results, since the mineral mixture contained, in addition to magnesium about 70 other trace minerals.”

7. Lines 251-253 - "...and excess of zinc can cause adverse effects, such as intestinal microbiota dysbiosis that will exacerbate a Clostridium difficile infection" - Reference missing

This was changed as requested. The sentence now reads: “Zinc deficiency has been associated with the development of diarrhea [8], and excess of zinc can cause adverse effects, such as intestinal microbiota dysbiosis that will exacerbate a Clostridium difficile infection [9; Zackular JP, Moore JL, Jordan AT, Juttukonda LJ, Noto MJ, Nicholson MR, et al. Dietary zinc alters the microbiota and decreases resistance to Clostridium difficile infection. Nat Med. 2016;22(11):1330-4.].”

8. Line 258 - 'fed efficiency' - e missing (feed efficiency).

The error was corrected. The sentence now reads: “Recently, CF231 taxa was associated with improved feed efficiency in pigs”

Reviewer 2 Report

The study under review results yielded more questions than answers.

To interpret the results, the following questions must be answered.

  1. How did the authors define Mg-deficient, Mg-control and Mg-High diet? Is the definition arbitrary?
  2. Why did the authors euthanatize the animal after 2 weeks, why not three weeks, why not more? 

By answering these 2 questions, we might be able to compare the study’s results to those published earlier. Can differences in the amount and the duration of Mg supplementation explain the discrepant results?

  1. Can H-Mg= 6000mg/kg diet induce diarrhoea itself? If yes, it might be the diarrhoea that induced microbiota changes, irrespectively of the aetiology
  2. Although low dietary magnesium consumption was associated in this study with a microbiota with a higher capacity to harvest energy from the diet did the authors detected this outcome in the rats? Is this reflected in Table 1?

Minor comment.

Krustal Wallis test, p<0.05 under Table 1 is irrelevant 

Author Response

Reviewer: 2

We thank the reviewers for their valuable comments and suggestions, which have strengthened the revised manuscript substantially. Our point-by-point response to their concerns is provided below.

  1. How did the authors define Mg-deficient, Mg-control and Mg-High diet? Is the definition arbitrary?

In the current literature the diet magnesium content management is referred as low (8-330 mg/kg), control (510-2430 mg/kg), and high (2000-6000 mg/kg).

Low diet:

  • Kumar BP and Shivakumar K. Depressed antioxidant defense in rat heart in experimental magnesium deficiency. Implications for the pathogenesis of myocardial lesions. Biol Trace Elem Res 1997;60(1-2):139-144. doi: 10.1007/BF02783317. (8 mg/kg).
  • Rayssiguier Y et al. Effect of magnesium deficiency on lipid metabolism in rats fed a high carbohydrate diet. J Nutr. 1981;111(11):1876-1883. doi: 10.1093/jn/111.11.1876. (35 mg/kg).
  • Shah NC et al. Short-term magnesium deficiency downregulates telomerase, upregulates neutral sphingomyelinase and induces oxidative DNA damage in cardiovascular tissues: relevance to atherogenesis, cardiovascular diseases and aging. Int J Clin Exp Med 2014;7(3):497-514. eCollection 2014. (60 mg/kg).
  • Fonseca FA et al. Dietary magnesium improves endothelial dependent relaxation of balloon injured arteries in rats. Atherosclerosis 1998;139(2):237-242. doi: 10.1016/s0021-9150(98)00069-0. (330 mg/kg).

Control diet:

  • Kumar BP and Shivakumar K. Depressed antioxidant defense in rat heart in experimental magnesium deficiency. Implications for the pathogenesis of myocardial lesions. Biol Trace Elem Res 1997;60(1-2):139-144. doi: 10.1007/BF02783317. (510 mg/kg).
  • Rayssiguier Y, Gueux E, Weiser D. Effect of magnesium deficiency on lipid metabolism in rats fed a high carbohydrate diet. J Nutr. 1981;111(11):1876-1883. doi: 10.1093/jn/111.11.1876. (1000 mg/kg).
  • Weglicki WB et al. Antioxidants and the cardiomyopathy of Mg-deficiency. Am J Cardiovasc Pathol. 1992;4(3):210-215. (2430 mg/kg).

High diet:

  • Fonseca FA et al. Dietary magnesium improves endothelial dependent relaxation of balloon injured arteries in rats. Atherosclerosis 1998;139(2):237-242. doi: 10.1016/s0021-9150(98)00069-0. (2000 mg/kg).
  • Blache D et al. Long-term moderate magnesium-deficient diet shows relationships between blood pressure, inflammation and oxidant stress defense in aging rats. Free Radic Biol Med 2006 ;41(2):277-284. doi: (10.1016/j.freeradbiomed.2006.04.008. Epub 2006 Apr 21. (3200 mg/kg).
  • Pere AK et al. Dietary potassium and magnesium supplementation in cyclosporine-induced hypertension and nephrotoxicity. Kidney Int 2000;58(6):2462-2472. doi: 10.1046/j.1523-1755.2000.00429.x. (6000 mg/kg).

According to above information of magnesium content we used as deficient diet 60 mg/kg, as control diet 1000 mg/kg, and high diet 6000 mg/kg.

2. Why did the authors euthanatize the animal after 2 weeks, why not three weeks, why not more?

There are works that report changes when variable amounts of magnesium are administered for different intervals of time. These intervals range from 1 week to 2 years.

  • Rayssiguier Y et al. Effect of magnesium deficiency on lipid metabolism in rats fed a high carbohydrate diet. J Nutr 1981;111(11):1876-1883. doi: 10.1093/jn/111.11.1876. (Treatment for 1 week).
  • Weglicki WB et al. Antioxidants and the cardiomyopathy of Mg-deficiency. Am J Cardiovasc Pathol 1992;4(3):210-215. (Treatment for 2-3 weeks).
  • Shah NC et al. Short-term magnesium deficiency downregulates telomerase, upregulates neutral sphingomyelinase and induces oxidative DNA damage in cardiovascular tissues: relevance to atherogenesis, cardiovascular diseases and aging. Int J Clin Exp Med 2014;7(3):497-514. eCollection 2014. (Treatment for 3 weeks).
  • Hélène Martin H et al. Effects of long-term dietary intake of magnesium on oxidative stress, apoptosis and ageing in rat liver. Magnes Res 2008;21(2):124-30. (Treatment for 2 years).

In particular, Rayssiguier et al. (1981) observed induced-magnesium deficiency detrimental changes after 8 days in the content of serum triglycerides and cholesterol in very low-density lipoprotein (VLDL), low-density lipoprotein (LDL) and high-density lipoprotein (HDL) fractions. Then, under the consideration that magnesium deficiency has effects in about one week, and the fact that rats developed diarrhoea as consequence of magnesium high diet treatment, the rats were euthanized after 2 weeks of treatment.

3. Can differences in the amount and the duration of Mg supplementation explain the discrepant results?

We thank the reviewer for this comment. The discrepant results could be related to the hypervariable region that was analyzed (V3; Jørgensen. Dietary magnesium deficiency affects gut microbiota and anxiety-like behavior in C57BL / 6N mice. Acta Neuropsychiatrica 2015. DOI: 10.1017 / neu.2015.10; Winther. Dietary magnesium deficiency, alters gut microbiota and leads to depressive-like behavior. Acta Neuropsychiatrica 2015. DOI: 10.1017 / neu.2015.7) or V3-V4;Crowley EK. Dietary Supplementation with a Magnesium-Rich Marine Mineral Blend Enhances the Diversity of Gastrointestinal Microbiota. Mar Drugs. 2018;16(6)). Also, the methods to evaluate intestinal microbiota; denaturation gradient gel electrophoresis (Jørgensen. Dietary magnesium deficiency affects gut microbiota and anxiety-like behavior in C57BL / 6N mice. Acta Neuropsychiatrica 2015. DOI: 10.1017 / neu.2015.10; Winther. Dietary magnesium deficiency alters gut microbiota and leads to depressive-like behavior. Acta Neuropsychiatrica 2015. DOI: 10.1017 / neu.2015.7) or Real-time quantitative PCR for microbial cecal content (Pachikian, Changes in Intestinal Bifidobacteria Levels Are Associated with the Inflammatory Response in Magnesium-Deficient Mice. J. Nutr. 140: 509–514, 2010) Otherwise, Winther et al., did not use any control group for their comparisons. Moreover, the author who studies the same hypervariable region and performs massive sequencing, Crowley et al., suggest being cautious with the results since the mineral mixture contained, in addition to magnesium, other trace minerals (Crowley EK. Dietary Supplementation with a Magnesium-Rich Marine Mineral Blend Enhances the Diversity of Gastrointestinal Microbiota. Mar Drugs. 2018;16(6)).

4. Can H-Mg= 6000mg/kg diet induce diarrhoea itself? If yes, it might be the diarrhoea that induced microbiota changes, irrespectively of the aetiology

Although oral Mg has been reported to cause diarrhea in clinical studies (al-Ghamdi et al., 1994; Barragán-Rodríguez et al., 2008), the experiments listed above did not report diarrhea in animal models fed Mg supplementation diets [5,6,8]. We only know about one report that evaluated the effect of Mg-induced diarrhea in the gut microbiota, which used “purgative doses” of 0.5 mL saturated MgSO4 solutions (probably 129 mg of total Mg). (Phillips et al., 1978). In our case, the average Mg supplemented diet consumption (15.7 g/rat/day) did not quite reach the Mg quantity used for the purgative dose (94.7 mg of total Mg) but it is probable that diarrhea may as well have produce a change in the microbiota as reported in the previous article (Phillips et al., 1978).

References:

  • Al-Ghamdi S, Cameron E, Sutton R. Magnesium Deficiency: Pathophysiologic and Clinical Overview 1994;24(5):737-752
  • Barragán-Rodríguez L, Rodríguez-Morán F, Guerrero-Romero F. Efficacy and safety of oral magnesium supplementation in the treatment of depression in the elderly with type 2 diabetes: a randomized, equivalent trial 2008;21(4):218-223
  • Phillips M, Adrian L, Leach WD. The mucosa-associated microflora of the rat intestine: a study of normal distribution and magnesium sulphate induced diarrhea 1978;56(6):649-662

5. Although low dietary magnesium consumption was associated in this study with a microbiota with a higher capacity to harvest energy from the diet did the authors detected this outcome in the rats? Is this reflected in Table 1?

We thank the reviewer for this comment. Paragraphs addressing this have been included in the Discussion section. “Several studies have reported that lower magnesium consumption is correlated with an increased risk of insulin resistance [24-26]. It is possible that in the long term, the taxa founded in the L-Mg group favor the development of obesity and insulin resistance. To confirm this observation, it is recommended that the intervention duration be longer and/or lower doses of magnesium be used.”

Minor comment.

  1. Krustal Wallis test, p<0.05 under Table 1 is irrelevant

We thank the reviewer for this comment. The KW test was to indicate that no significant differences were shown in the evaluated parameters.

Round 2

Reviewer 2 Report

Authors provided responses to my concerns. However, the text has not been amended to include the responses to 4 out of 5 concerns. Please amend it accordingly.

Moreover, significant amount of information regarding the methods was deleted in the revised document. This info is important to judge the methodology of the experiments and should be presented in the text or as supplemental file.

One new author was included. Did the corresponding author provide the reasons of this entry? Did all authors approve this entry?

Author Response

The authors provided responses to my concerns. However, the text has not been amended to include the responses to 4 out of 5 concerns. Please amend it accordingly.

R. This was done as requested, corresponding revisions in the revised manuscript are shown in yellow. The sentences now reads: 

-        The Animal Committee approved the study of the Chemistry Faculty, UNAM, Mexico City on 09/20/2019 (CICUAL/384-2/19). Animals were obtained from Medicine Faculty, UNAM. Male Wistar rats aged 7 weeks were kept in individual cages with controlled room temperature with free access to water and diet and 12-h light-dark cycles. Rats were random assignment into three groups; six rats were fed a control diet (C-Mg; 1,000 mg/kg), six rats were fed a low magnesium content diet (L-Mg; 60 mg/kg), and five rats were fed a high magnesium content diet (H-Mg; 6,000 mg/kg) for two weeks. The rats were euthanized after 2 weeks of treatment by the consideration that magnesium deficiency has effects in about one week [12], and the fact that rats developed diarrhea as a consequence of magnesium high diet treatment.

-        "A mineral mix without MgO (TD 180705) were used to modulate magnesium concentration by adding MgO (Sigma 243388) to 60, 1000 or 6000 mg/kg based on the experiments of Rayssiguier et al., [12] and Martin et al., 2008 [13]."

-        “Interestingly, the present study demonstrates that dietary magnesium content affects the structure, composition, and functional pathways of rats' intestinal microbiota in two weeks. Mainly, high dietary magnesium decreased community diversity, while low dietary magnesium did not modify diversity. Our results contrast with previous research reported by Winther et al., who demonstrated that a diet deficient in magnesium modified bacterial diversity in mice [16]. However, this study compared magnesium deficiency with a magnesium excess, lacking a group of control magnesium content. Moreover, the discrepant results could be related to the hypervariable region that was analyzed [V3 or V3-V4] [4, 16, 17]. Also, the methods to evaluate intestinal microbiota; denaturation gradient gel electrophoresis [4, 16], real-time quantitative PCR [18] or sequencing [4]. Additionally, the author who studies the same hypervariable region and performs massive sequencing as Crowley et al., suggest being cautious with the results since the mineral mixture contained, in addition to magnesium, other trace minerals [17]. Moreover, the lower diversity observed with high dietary magnesium has also been observed in other minerals such as zinc and iron. Zinc deficiency has been associated with the development of diarrhea [8], and excess of zinc can cause adverse effects, such as intestinal microbiota dysbiosis that will exacerbate a Clostridium difficile infection [9]. Regarding iron, iron-supplemented mice show less diversity than the iron-deficient group after 27 days [19]"

-        "It is important to point out that, in the present study, the H-Mg group presented diarrhea without affecting weight gain or animal growth. Although oral Mg has been reported to cause diarrhea in clinical studies [28, 29], the experiments listed above did not report diarrhea in animal models fed Mg supplementation diets [12, 13]. We only know about one report that evaluated the effect of Mg-induced diarrhea in the gut microbiota, which used “purgative doses” of 0.5 mL saturated MgSO4 solutions [129 mg of total Mg]. In our case, the average Mg supplemented diet consumption [15.7 g/rat/day] did not quite reach the Mg quantity used for the purgative dose [94.7 mg of total Mg] but it is probable that diarrhea may as well have produced a change in the microbiota as reported in the previous article [30]. 

Moreover, a significant amount of information regarding the methods was deleted in the revised document. This info is important to judge the methodology of the experiments and should be presented in the text or as a supplemental file.

R. This was done as requested. The methods of Bacterial DNA purification, amplification, and sequencing were added in the supplemental material. 

One new author was included. Did the corresponding author provide the reasons of this entry? Did all authors approve this entry?

R. Dr. Barrera-Oviedo contributes to metabolic chambers to obtain feces samples without urine contamination. All the authors signed the "Authorship Change Form". The form was attached and uploaded with the first resubmission
